# Comparison of PCR Techniques in Adulteration Identification of Dairy Products

**Baiyi Li [1,†], Mingxue Yu [1,†], Weiping Xu [1], Lu Chen [1,*] and Juan Han [1,2,3,*]**

1 Institute of Food and Nutrition Development, Ministry of Agriculture and Rural Affairs, Beijing 100081, China; libaiy1@126.com (B.L.); mingeryu0228@163.com (M.Y.); xuweiping@caas.cn (W.X.)

2 Laboratory of Safety & Nutritional Function Risk Assessment for Agricultural Products of China Ministry of Agriculture and Rural Affairs, Beijing 100081, China

3 Digital Agriculture and Rural Research Institute of CAAS (Zibo), Zibo 255022, China

* Correspondence: chenl@mail.hzau.edu.cn (L.C.); hanjuan@caas.cn (J.H.);
  Tel.: +86-10-82106427 (J.H.); Fax: +86-10-82105184 (J.H.)

† These authors contributed equally to this work.

**Abstract:** Economic profit-driven food adulteration has become widespread in the dairy industry. One of the most common forms of dairy adulteration is the substitution of low-priced milk for high-priced milk. This has prompted regulatory authorities to focus on various means of authenticity testing. So far, many methods have been developed. Since milk adulteration has been upgraded, which has forced the testing methods to meet the needs of detection, which include DNA-based PCR methods. PCR and PCR-derived methods exhibit multiple advantages for authenticity testing, such as high stability, fast speed, and high efficiency, which meet the needs of modern testing. Therefore, it is important to develop rapid, reliable, and inexpensive PCR-based assays for dairy adulteration identification. In order to provide perspectives for improving adulteration identification methods, this review first summarizes the DNA extraction methods, then compares the advantages and disadvantages of various PCR authenticity testing methods, and finally proposes the directions for improving dairy product adulteration identification methods.

**Keywords:** PCR; adulteration identification method; dairy products; quality control; food fraud





## 1. Introduction

Food fraud has become a worldwide issue [1], resulting in huge economic losses and health risks [2]. Dairy products, which are made of essential nutrients, are the ideal food for the majority of humans, and thus, a huge consumer market and an adulteration epicenter have been formed [3,4]. With the enhanced seriousness of dairy product adulteration, the requirement for cheap, sensitive, and efficient methods for adulteration tests has been significantly increased. The adulteration of dairy induction comes in diverse forms such as the abuse of additives, the addition of milk extracts, the dilution with water, and the mixture of low-priced milk [5]. One common form of dairy product adulteration is the addition of low-priced milk to high-priced milk. Gonçalves et al. (2012) identified the source of commercial dairy products in Porto, and the results showed that 12.5% (12/96) of samples failed to confirm the description contents [6]. Similarly, Di Pinto et al. (2017) tested the authenticity of 80 goat cheeses from the Italian market and found that 80% were adulterated with cow/sheep milk [7]. Furthermore, Zhang et al. (2022) detected the source of 46 commercial minor species' milk from the Chinese market, in which 15 samples were species false labeling [8]. Such adulteration can put the health and even the life of the consumer at risk, since any milk protein of unknown origin, especially cow milk protein, is a potential allergen [9–11]. Based on this, adulteration identification aimed at species-specific traits and quantitative estimation of adulteration levels are attractive to the supervisory authority.

In the past decades, numerous species-specific adulteration identification methods were developed based on the characteristic substances such as proteins, lipid acids, sugars, and DNA in dairy products. Protein-based identification tools, such as isoelectric-focusing, chromatography, and immunochemical technologies, exhibit advantages such as being fast, efficient, and affordable, and they are considered to be appropriate for species identification in dairy products [12–14]. However, instability of the protein structure tends to result in identification bias, limiting the application of protein-based identification in processed dairy products [12,13]. Although the mass spectrometry (MS) method allows accurate species identification in processed dairy products using marker peptides, the requirements for professional equipment and personnel significantly restrict its wide application [13,14]. In addition, many spectroscopic methods targeting at a feature group of nutrients (mainly including lipid acid, protein, and carbohydrate) have been developed for dairy product adulteration identification, such as mid-infrared (MIR), near-infrared (NIR), front-face fluorescence spectroscopy (FFFS), Fourier-transform infrared (FT-IR), and nuclear magnetic resonance (NMR) [13,14]. Although spectroscopic methods are more suitable for daily monitoring than other methods, the exorbitant equipment costs restrict their application range [8]. With the development of polymerase chain reaction (PCR), many techniques for evaluating authenticity by estimating the residual DNA sequence in dairy products have emerged and are extensively used in the dairy industry [15,16].

The thermal stability of DNA is higher than that of other nutrients in dairy products, and more importantly, the detection of DNA is independent from immune responses [12,13]. These advantages have encouraged the development of PCR-based assays, and a variety of PCR assay forms have been derived, which play a key role in dairy industry adulteration identification (Figure 1). The quality of the DNA extraction is closely related to the successful implementation of these PCR-based methods [17]. The composition of dairy products greatly affects the extraction of DNA, and accordingly affects the subsequent amplification and testing [18]. Therefore, the development of appropriate DNA extraction methods is essential for the promotion and improvement of PCR-based methods. Considering this, the current review first summarizes DNA extraction methods from dairy products in great detail, and then analyses the advantages and disadvantages of PCR-based methods. This review provides new perspectives for developing sensitive, rapid, and inexpensive PCR-based methods for dairy product adulteration identification.

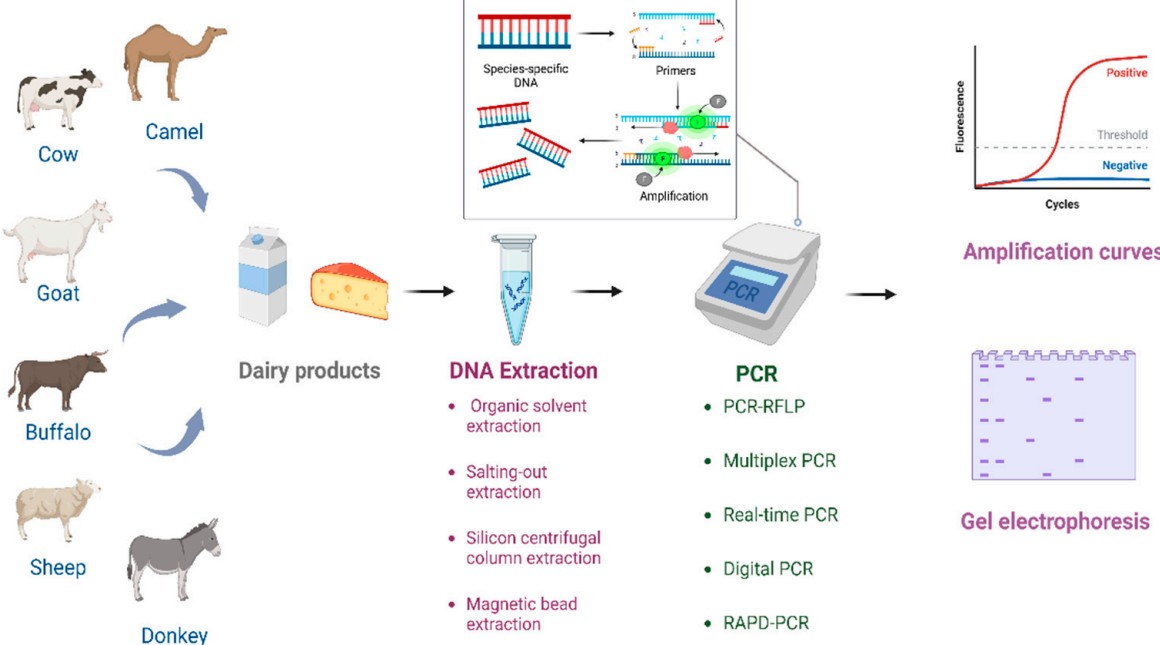

**Figure 1.** The process of identifying the authenticity of dairy products using PCR technology.

## 2. DNA Extraction Methods

The quality and quantity of the DNA extracted from dairy production is the key factor to the success of PCR methods used in dairy adulteration identification. Many factors, such as somatic cell number, milk composition, and DNA extraction methods, can affect the quality and quantity of the DNA used in PCR methods, and in turn, affect the adulteration detection effectiveness [19,20]. Many factors might affect the concentration of somatic cells in dairy products such as milk-producing animal breeds, lactation stage, and health as well as environments [21,22]. The components of dairy products, such as fat, impact the efficiency of DNA extraction [23]. In addition, the DNA isolated from microorganisms in dairy products can dilute the target DNA concentration [24]. Therefore, an appropriate method is essential to extract DNA from different dairy products. Many DNA extraction methods based on organic solvents, salting-out, silicon centrifugal columns, and magnetic beads have been developed to accommodate different types of dairy products.

### 2.1. Organic Solvent Extraction Method

Organic solvent extraction is the traditional method of DNA extraction. This method first dissociates cells using surfactants such as hexadyltrimethyl ammonium bromide (CTAB) and sodium dodecyl sulfate (SDS) and then releases DNA with protease K and ethylenediaminetetraacetic acid (EDTA) buffer. The proteins and other cellular components are separated from the DNA in the water phase with the phenol/chloroform/isoamyl alcohol mixture by centrifugation. Finally, the DNA is precipitated by isopropanol, washed with 70% ethyl alcohol, and then resuspended in Tris-ethylenediaminetetraacetic acid (TE) buffer [25]. Compared with that isolated by other extraction methods, the DNA isolated by the organic solvent extraction method exhibits high integrity, quality, and yield [26]. However, this method also has obvious shortcomings such as being time-consuming, use of toxic solvents, and poor reproducibility, limiting its large-scale applications [27].

### 2.2. Salting-Out Extraction Method

The salting-out extraction method separates DNA from proteins and other impurities with an appropriate concentration of sodium chloride solution by taking advantage of the solubility differences among DNA, proteins, and other impurities in NaCl solution. This method is less labor-intensive, requires no toxic reagents, and exhibits high efficiency [28]. However, the DNA extracted by the salting-out method displays lower purity than that extracted by the organic solvent method [12,29]. Compared to silicon centrifugal columns, the salting-out extraction method needs more raw material and yields less DNA, exhibiting a smaller application range [30].

### 2.3. Silicon Centrifugal Column Extraction Method

The cell lysis buffer used for silicon centrifugal column extraction method contains chaotropic agents such as guanidinium chloride, which bring about the DNA binding to silicon column selectively and reversibly [31]. Subsequently, the DNA is washed with an alcohol solvent and eluted with pure water/TE buffer. The DNA extraction rate from different types of dairy products depends on the lysis buffer type and volume. The silicon centrifugal column extraction method has been widely used for DNA extraction. Although the DNA yield extracted by this method is lower than that by organic reagent extraction, this method exhibits higher consistency, higher efficiency, and less time-consuming advantages [29].

### 2.4. Magnetic Bead Extraction Method

The magnetic beads are coated with a layer of carboxylate, which can specifically absorb DNA from cell lysates through the ionic bridges formed by the negatively charged phosphate group, carboxylate, and the dissociated salt ions. Once the polyethylene glycol (PEG) and salts are removed, the added aqueous molecules rapidly and sufficiently hydrate the DNA, and hydrated DNA abolishes the ionic interactions among the above-mentioned

three components, allowing the DNA adsorbed on the magnetic beads to be purified [32,33]. The magnetic bead extraction method is the most suitable method for high-throughput DNA extraction compared with other methods. However, comparative studies indicate that both the quantity and quality of DNA extracted by the magnetic bead method are lower than those extracted by silicon centrifugal columns extraction method [34].

With the rapid development of molecular biology and medical technology, the DNA extraction technology has advanced towards the direction of being simple, fast, high-quality, high-purity, and high-throughput so as to meet the requirements of large sample size and high-quality experiments. The current common DNA extraction methods have their own advantages, disadvantages, and application scope, which are summarized in this review (Table 1).

**Table 1.** Summary of advantages and disadvantages of various DNA extraction methods.

| Extraction Method | Advantages | Disadvantages | Refs. |
|---|---|---|---|
| Organic solvent extraction method | High purity; Large quantity; Intact fragments; Low cost | Use of toxic reagents; Poor reproducibility; Time-consuming | [25,26,35] |
| Salting-out extraction method | Simplicity; No use of toxic reagents; Highly efficient | Low yield; Less application range; Large quantities of raw material | [12,28,29] |
| Silicon centrifugal columns extraction method | Higher integrity and purity; Easy operation; Less time consumption | More pronounced shearing of genomic DNA; Higher cost; Lower concentration of DNA | [28,36,37] |
| Magnetic beads extraction method | Less time consumption; Easy operation; Efficient | Low yield | [33,34,36] |

## 3. PCR-Based Dairy Product Authenticity Testing

PCR is widely used for animal origin identification in the dairy industry. The different species-specific DNA targets can be amplified by PCR, separated by agarose gel according to the fragment size, and visualized by DNA dye [38,39]. PCR has multiple advantages; for example, it is simple, less time-consuming, and it does not rely on expensive instruments [40]. DNA sequences with high copy numbers and high species specificity are used as targets for PCR assays. Mitochondrial DNAs with high inter-species specificity and high intra-species conservation are usually used as targets for PCR assays. Meanwhile, some nuclear DNA can also be used as targets for PCR assays. Recently, more and more derivative methods have been developed to improve PCR techniques for dairy product adulteration identification [12–14] such as PCR-restriction fragment length polymorphism (PCR-RFLP), multiplex PCR, real-time PCR, and digital PCR (dPCR) based on known DNA target sequences as well as random amplified polymorphic DNA-PCR (RAPD-PCR) developed based on unknown target sequences.

### 3.1. PCR-RFLP

RFLP-PCR is a simple, fast, and inexpensive method. It starts with sequence amplification by PCR, followed by enzymatic digestion with a restriction endonuclease, conducts separation by electrophoresis, and ends up with visualization by a UV gel imager. This method was first proposed by Plath et al. in 1997 to identify cow milk from sheep milk, goat milk, and cheese with polyacrylamide gel electrophoresis followed by PCR-RFLP [41]. Then, Abdel-Rahman et al. (2007) used PCR-RFLP and agarose gel to identify buffalo milk, cow milk, and sheep milk with the cytochrome b (*cyt b*)gene as the DNA target [42]. With the aid of computational biology, Lanzilao et al. (2005) set up an enzyme mixture (HaeIII, TaqI, and MwoI) acting on the *cyt b* gene to identify the milk sources from cow, sheep, goats, and buffaloes [43]. Then, Vafin et al. (2022) mixed XbaI and PsiI to digest the κ-casein gene and generate the RFLP profiles, enabling the identification of the milk sources (sheep, goat, cow, and buffalo) of dairy products [44].

### 3.2. Multiplex PCR

The multiplex PCR (mPCR) technique has been developed to increase sensitivity and efficiency. The mPCR can amplify multiple targets in a single PCR reaction to identify the multi-sourced dairy products with the advantages of high efficiency, high throughput, and inexpensiveness. This method was first developed by Bottero et al. (2003) to simultaneously detect multiple animal sources in dairy products with a detection limit of 0.5%. Multiplex PCR was used for the detection of laboratory-made cheeses (from cow milk, goat milk, and sheep milk) and commercial Italian cheeses with mitochondrial 12S and 16S rRNA genes as the targets [45]. Tsirigoti et al. (2020) designed a primer mixture with two *cyt b* genes based on cow and sheep and one DNA sequence based on the goat D-loop range to identify the milk sources, exhibiting a detection limit of 0.1% [46]. With the advancements in molecular biology technology, many quantitative PCR-based multiplex PCRs have been developed for the detection of dairy products.

### 3.3. Real-Time PCR

Real-time PCR is also known as quantitative real-time PCR, and it can monitor the amplification products in each PCR cycle using fluorescent reporter molecules. This real-time PCR method is gradually replacing traditional PCR due to its excellent sensitivity, specificity, and high consistency, as well as qualitative and quantitative properties. Recently, the qPCR has been classified into two types based on the sequence detection system. In one type, DNA is detected by double-stranded DNA-embedded dye represented by SYBR Green, and in the other type, DNA is detected by fluorophore-labelled oligonucleotides (represented by TaqMan). These two types of qPCR have their own advantages and different application scenarios. The qPCR using DNA dye has the advantage of being economical and rapid, but it has its own disadvantages such as low specificity and the requirement for a lysis curve analysis after amplification. In contrast, the qPCR using fluorophore-labeled oligonucleotides has the advantages of high specificity and the ability to set multiple detection targets, thus expanding its application to dairy product adulteration detection.

The qPCR with SYBR Green was first used by Hebert et al. (2003) to quantify the ratio of cow milk to buffalo milk in mozzarella cheese with cytochrome oxidase subunit 1 (*coI*) as the target gene [47]. Lopez-Calleja et al. (2007) first reported that qPCR with TaqMan probes could identify goat milk or cow milk from sheep milk by amplifying the 12S rRNA sequence, with a linear dynamic range of 0.6–10% [48]. The target DNA and genes used in qPCR are usually the mitochondrial 12S rRNA gene and *cyt b* gene. Numerous qPCR-based derivative methods, such as multiplex primer design and high-resolution melting (HRM) analysis, have been developed to extend their application scope.

Cottenet et al. (2011) first designed two probes with *cyt b* genes from cow and buffalo as targets, making the detection of two milk sources possible [49]. Later, duplex PCR based on 16S rRNA and D-LOOP genes was used to detect adulterated milk sources (camel, horse, and goat) with a detection limit of 0.1% in raw milk and 0.5% in ultra-high-temperature (UHT)-sterilized milk [50]. The triple qPCR is developed to simultaneously detect the target 12S rRNA gene of the species to be detected and the endogenous control by TaqMan probes, based on which the simultaneous milk source detection of cow and mare, cow and goat, sheep and goat, and camel and cow has been successfully implemented [51–54]. Agrimonti et al. (2015) developed a quadruplex PCR targeting the *cyt b* gene and 12S rRNA to identify and quantify milk sources from cow, sheep, goat, and buffalo with a detection limit of 0.1% [55].

As a highly efficient and robust PCR technique, high-resolution melting (HRM) analysis is not limited by mutation sites or types, and it is used to analyze mutations, single nucleotide polymorphisms (SNPs), methylation, and mapping of samples without requiring sequence-specific probes [56]. Like many fluorescent PCR techniques, HRM inserts a specific dye into the double-stranded DNA to detect the samples by real-time monitoring of the binding of the fluorescent double-stranded DNA dye to the PCR amplification product and recording a high-resolution melting curve. Compared with conventional PCR methods,

HRM analysis is performed in the same tube as the PCR amplification step, and thus HRM requires no purification or separation of amplicons [57]. This makes HRM analysis faster, less laborious, and more suitable for high-throughput assays than other methods, thus increasing its applicability for food identification, especially for dairy product identification [58]. Ganopoulos et al. (2013) first reported that HRM analysis could be used to detect and quantify the cow milk in commercial samples of Feta cheeses (protected designation of origin, PDO), samples of sheep-goat cheeses (non-PDO), and samples of goat cheeses. In their report, the D-loop sequences of cattle and sheep and the tRNA-Leu gene of goats were used as targets to identify the authenticity of the label with a detection limit of 0.1% [59].

### 3.4. Digital PCR (dPCR)

To further reduce the detection limit, digital PCR (dPCR) has also been used to test the authenticity of dairy products. dPCR is an absolutely quantitative PCR, and it starts with the limited dilution of PCR reactants, followed by PCR amplification in different reaction chambers, and ends with qualification of initial copy number or concentration of target molecules based on the Poisson distribution principle and the number and proportion of positive microdroplets [60]. This method was developed in the late 1980s [61], and it was first named digital PCR in 1999 [62]. Compared to qPCR, dPCR has the following advantages: sensitivity up to the single-molecule level, detection limit as low as 0.001%, ability to absolutely quantify initial levels of target molecules without requiring standard curves, and independence from Ct value, PCR amplification efficiency, and PCR inhibitor effects [63]. In the past, dPCR was widely used in the dairy testing industry to detect pathogenic microbial contamination in dairy products, and at present, it is used to identify milk sources [64–67]. Cutarelli et al. (2021) used the droplet digital PCR (ddPCR) with *cyt b* gene as the target to identify the cow and buffalo milk in mozzarella cheese [64].

### 3.5. Randomly Amplified Polymorphic DNA PCR (RAPD-PCR)

Since all the above-mentioned PCR techniques require known gene sequences, the identification of the milk sources will be hampered when no known gene sequences are available. As a result, some PCR methods requiring no known sequences have been developed such as RAPD-PCR. This RAPD-PCR method was first proposed by Williams et al. in 1990 to infer the gene arrangement patterns and phenotypes by analyzing the polymorphism of PCR products [68]. RAPD-PCR has many advantages over other PCR methods, such as lower template requirements, independence from specific primers, ease of use, and less interfering factors [69]. Therefore, this method is usually used for the classification of breeds including the classification of microbial communities in dairy products [70,71]. Cunha et al. (2016, 2017) employed this method to detect fraudulent manufacturing of dairy products by combining the sequence-characterized amplified region [72,73].

The above PCR detection techniques are used for the detection of milk sources in dairy products. The detection targets and methods for these technologies are summarized in Table 2.

**Table 2.** Summary of PCR-based techniques for dairy products authenticity testing.

| Techniques | Species Identification | Types of Dairy Products | DNA Target | Detection Limit | Ref. |
|---|---|---|---|---|---|
| PCR-RFLP | Identification of the milk sources from cow, sheep, and buffalo milk | Raw milk | *cyt b* gene | Qualitative analysis | [42] |
| | Identification of the milk sources from cow, sheep, goat, and buffalo | Raw milk | *cyt b* gene | Qualitative analysis | [43] |
| | Identification of the milk sources from cow, sheep, and buffalo milk | Raw milk | *κ-casein* gene | Qualitative analysis | [44] |
| | Identification of the milk sources from three Egyptian goat breeds (Bariki, Damascus, and Zaraibl) | Blood sample | *myostatin* (*MSTN*) gene and *prolactin* (*PRL*) gene | Qualitative analysis | [74] |

**Table 2.** *Cont.*

| Techniques | Species Identification | Types of Dairy Products | DNA Target | Detection Limit | Ref. |
|---|---|---|---|---|---|
| Multiple PCR | Identification of the cheese sources from cow, sheep, and goat milk | Cheese | *Mitochondrial 12S* and *16S rRNA* genes. | 0.5% detection limit | [45] |
| | Identification of the milk sources from cow, goat, and sheep milk | Raw milk | *cyt b* genes (cow and sheep) D-loop range (goats) | 0.1% detection limit. | [46] |
| | Identification of the milk sources from camel, horse, and goat | Raw milk | *16S rRNA* gene and D-LOOP range | 0.1% detection limit. | [50] |
| | Identification of the milk sources from cow, goat, sheep, and buffalo milk | Raw milk | *cyt b* gene and *12S rRNA* | 0.1% detection limit. | [55] |
| | Identification of the milk sources from cow, sheep, and goat milk | Blood sample | *16S rRNA* (sheep) and *12S RNA* (cow, sheep, and goat) | 0.1% detection limit. | [75] |
| Real-time TaqMan PCR | Identification of goat milk or cow milk from sheep milk | Raw milk | *12S rRNA* | Linear dynamic range of 0.6–10% | [48] |
| | Identification of the milk sources from cow and mare milk | Raw milk | *12S rRNA* | 1 pg of cow DNA; 1 pg of mare DNA, | [51] |
| | Identification of the cheese, and milk sources from cow and goat milk | Cheese and raw milk | *12S rRNA* | 0.005 ng/μL in milk and 0.01 ng/μL in cheese | [52] |
| | Identification of the cheese and milk sources of sheep and goat milk | Cheese and raw milk | *12S rRNA* | 0.005 ng/μL in milk and 0.01 ng/μL in cheese | [53] |
| | Identification of the milk, yogurt, cheese, milk powder, and milk beverage sources from cow and camel milk | Milk, yogurt, cheese, milk powder, and milk beverage | *12S rRNA* | 0.0025 to 0.001 ng/μL (milk); 0.5 to 0.001 ng/μL (yogurt); 1 to 0.05 ng/μL (cheese); 0.01 ng/μL (milk powder); 0.001 ng/μL (milk beverage) | [54] |
| Real-time SYBR Green PCR | Identify the cheese source from cow and buffalo milk | Cheese | *Cytochrome oxidase subunit 1* (*coI*) | - [a] | [47] |
| | Identification of the milk sources from cow, and buffalo | Cheese | *Cytochrome oxidase subunit 1* (*coI*) | 0.5 ng/μL | [76] |
| | Identification of the milk sources from camel, cow, and goat | Milk, milk powder, and milk soap | *Follicle stimulating hormone receptor* (*FSHR*) gene (camel and goat) and 12S RNA (cow) | Rang of 0.001–0.002% | [77] |
| Digital PCR | Identify the cheese source from cow and buffalo milk | Cheese | *cyt b* genes | 0.1% detection limit. | [64] |
| RAPD-PCR | Identify the cheese source from four sheep breeds' milk | Cheese | Sequence-characterized amplified region | Qualitative analysis | [73] |

[a] not reported.

In summary, PCR and PCR-derived methods have been applied for the authenticity identification of dairy products. However, each of these methods has a different scope of application. In order to provide a resource for the subsequent development of rapid, accurate, and cost-effective identification methods, the advantages and disadvantages of the above PCR-based dairy product identification methods are summarized in Table 3.

**Table 3.** Summary of advantages and disadvantages of PCR techniques.

| PCR Techniques | Cost | Detect Time | Analysis Procedure | Other |
| --- | --- | --- | --- | --- |
| PCR-RFLP | Low | Fast | Not quantitative analysis. | Simple. |
| Multiple PCR | Low | Fast | Both qualitative and quantitative analyses. | High-throughput; False-positive result |
| Real-time PCR | Low | Fast | Both qualitative and quantitative analyses. | Excellent sensitivity; Specificity; High consistency; False-positive result |
| Digital PCR | High | Fast | Both qualitative and quantitative analyses. | Accurate; Highly reproducible; Narrow dynamic range |
| RAPD-PCR | Low | Fast | Not quantitative analysis. | Independent from Specific primers; Ease to use; Less interfering factors |

## 4. Factors Affecting PCR-Based Dairy Product Authenticity Testing

DNA is a stable molecular marker for food authenticity testing. Food fraud detection methods based on PCR have been emerging in the past decades. However, many factors upstream of PCR detection limit its application in dairy product authenticity testing.

### 4.1. Disruption of DNA Integrity by Food Processing

DNA integrity affects downstream PCR amplification, which in turn affects the accuracy of the assay. Most dairy products, such as butter, cream, cheese, and UHT-sterilized milk, are heat-processed to extend their shelf life, which can result in DNA degradation [78]. During the preparation of milk powder, denatured proteins can bind to DNA, thus reducing the purity of the extracted DNA [79]. In addition to the effect of heating, the ripening process of cheese can also reduce the integrity of DNA, thereby affecting DNA amplification detection [80]. The DNA of the microorganisms that play a role in cheese ripening also reduces the purity of the milk DNA, hence influencing the PCR detection [81].

### 4.2. PCR Inhibitors

Many components that inhibit the PCR reaction are found in dairy products. For example, the fat and protease in cheese and butter, and even certain metal ions, such as calcium ions in milk, are PCR inhibitors [67,82–84]. Furthermore, PCR inhibitors, such as EDTA, polysaccharides, and other organic solvents, may be introduced during the DNA extraction process [85].

## 5. Conclusions and Future Perspectives

With the development of technology and improvements in people's health awareness, food regulations are becoming increasingly standardized and stringent. However, with the growth of the food supply chain, many global food frauds have resurfaced and spread, among which adulteration of dairy products has been the main issue of food fraud, which has made the authenticity of dairy products highly valued by the industry, market, and regulatory authorities. PCR and PCR-derived technologies are highly sensitive and reproducible and are well suited to the current needs of dairy product adulteration identification, which makes the development of PCR-based detection technologies and their application to dairy authenticity testing an area of great interest. Therefore, this paper reviews the current applications of PCR in dairy product adulteration identification and summarizes the advantages and disadvantages of each method in Table 3. This is of great importance for promoting the application of PCR technology in the authenticity of dairy products.

The quality of DNA extraction is critical for PCR detection and affects its accuracy. Therefore, future studies can be devoted to improving DNA extraction so as to avoid the effects of processing method, milk product composition, and extraction solvent on subsequent PCR amplification and analysis. Future studies are also suggested to focus on the development of more methods to increase the tolerance of DNA quality and widen

the application of PCR-based authenticity testing. Whole Genome Amplification (WGA) has become a valuable tool to overcome the problem of limited or insufficient amounts of DNA for downstream analysis. WGA can effectively improve the quality and quantity of DNA and can be successfully used for genomic analysis. Despite the lack of reports on the use of WGA for dairy product authenticity testing, WGA is a promising method to improve the sensitivity and accuracy of PCR-based authenticity detection. Therefore, more studies on WGA are needed to improve the application range of PCR for dairy product identification. dPCR is a PCR-based method developed as an alternative method to qPCR, and it is commonly used for allele detection. This method is unaffected by PCR inhibitors, and it can increase detection accuracy. dPCR provides a new solution to PCR-based dairy product authentication detection problems. Recently, there have been only a few detection methods based on dPCR used for dairy product authenticity testing, but dPCR exhibits a wide range of potential applications. Therefore, dPCR represents a new development direction for PCR-based adulteration identification.

**Author Contributions:** Conceptualization, J.H.; methodology, B.L. and M.Y.; investigation, B.L. and W.X.; resources, B.L. and J.H.; writing—original draft preparation, B.L., M.Y. and L.C.; writing—review and editing, W.X., L.C. and J.H.; supervision, J.H.; funding acquisition, J.H. All authors have read and agreed to the published version of the manuscript.

**Funding:** This research was funded by the Agricultural Science and Technology Innovation Program (CAAS-ASTIP-2023-IFND).

**Institutional Review Board Statement:** Not applicable.

**Data Availability Statement:** Data sharing not applicable.

**Conflicts of Interest:** The authors declare no conflict of interest.

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
