# Peer review of "Comparison of PCR Techniques in Adulteration Identification of Dairy Products"

_agriculture, doi:10.3390/agriculture13071450_

Round 1

Reviewer 1 Report

The manuscript Comparison of PCR techniques in adulteration identification of dairy products, addresses a very interesting and important topic related to adulteration of dairy products, which are an important part of the human diet. 

The pursuit of detection and prevention of adulteration should be a key element in improving food safety.

The manuscript was prepared interestingly describing quite extensively the possibility of using methods to detect adulteration.

In my opinion, Table 3 should be moved from the Conclusion and future perspectives section of the manuscript and placed in a new summary section.

I would like to draw attention to the incorrect citation of item number 5 by Wu et al. (2020), item number 9 by Mafra et al. (2022), item number 13 by Lipkin et al. (1993), item number 16 and 17, item number 20 by Kovacević et al. (2016), item number 59 by Cunha (2017), and item number 64 by Cordea and Mihaiu.

Reviewer 2 Report

References to 11 publications were not found. Authors must supply this or explain their inclusion and provide exact references. After that, it is necessary to carry out the inspection again.

References

322

Nr. 5 – not found

331

Nr. 10 – not found

343

Nr. 16 – not found

345

Nr. 17 – not exactly

354

Nr. 21 – not found

357

Nr. 22 – not found

359

Nr. 23 – not found

364

Nr. 27 – not found

368

Nr. 29 – not found 2019

378

Nr. 33 – not found

444

Nr. 64 – not found

Reviewer 3 Report

P1L39: aulteration: should be: adulteration

P1L40: characteritic: should be: characteristic

P1L44: Instablility: should be: instability

P3L107: reversiblely: should be: reversibly

P5L217: especially to dairy product identification: should be: especially for dairy product identification

P5L228: with quanlification: should be: with qualification

Minor editing of English language required

Reviewer 4 Report

Comments and Suggestions for Authors

1. The manuscript covers the use of PCR techniques in adulteration identification of dairy products. The problem itself is important, however, I am not sure if PCR techniques are suitable in this field. The authors did not prove that this approach is important in this field. Surprisingly, they stated that there are already better methods such as spectroscopic methods. Why should use PCR for adulteration identification? Still, I don’t know.

2. The structure of this review is very chaotic. It should be more organized. The sentences like (similar to): One common form of dairy product adulteration is the addition of low-priced milk to high-priced milk, which is rather harmful to health since it might bring in uncertain allergens. Author should mention the example of low priced milk and example of high priced milk. And explain them more details.

3. Line 76, the sentence can be improved. Add more information

4. For Table 1 and 3, author can add more data about the sample that used for extraction methods and for PCR techniques, respectively.

5. There is no a single figure that could explain in more precise way about PCR techniques in adulteration identification of dairy products.

6. Please re-write abstract and conclusion in accurate way

7. In this manuscript, there are a lot of mistakes. Some of them are related to poor English, some to a misunderstanding of the topic.

Considering all the above-mentioned issues as well as the general chaotic structure of the manuscript I recommend resubmit after major revision this manuscript for the publication in Agriculture journal.

Round 2

Reviewer 4 Report

In my assessment, the author's response to the reviewer' concerns was inadequate, particularly in light of the opinions provided by expert regarding the major repairs. Furthermore, the author neglected to include a figure to support the PCR techniques in adulteration identification of dairy products, and the scientific significance of the study was not sufficiently prominent. Considering these factors, I have concluded that the manuscript was not suitable for publication in Agriculture. Consequently, the manuscript was rejected due to its failure to meet the journal's requirements for novelty and impact.
